# Non-Celiac Wheat Gluten Sensitivity Model: Effects on Hepatic Morphophysiology of Wistar Rats

**DOI:** 10.3390/nu17111842

**Published:** 2025-05-28

**Authors:** Ana Luiza Russo Duarte, Gabriela Barone Volce da Silva, Anne Caroline Santa Rosa, Ghiovani Zanzotti Raniero, Antonio Roberto Giriboni Monteiro, Gustavo Henrique de Souza, Anacharis Babeto de Sá-Nakanishi, Jurandir Fernando Comar, Roberto Kenji Nakamura Cuman, Maria Raquel Marçal Natali

**Affiliations:** 1Graduate Program in Biological Sciences, State University of Maringa, Maringa 87020-900, Brazil; 2Department of Food Engineering, State University of Maringa, Maringa 87020-900, Brazil; 3Department of Biochemistry, State University of Maringa, Maringa 87020-900, Brazil; 4Department of Pharmacology and Therapeutics, State University of Maringa, Maringa 87020-900, Brazil; 5Department of Morphological Sciences, State University of Maringa, Maringa 87020-900, Brazil

**Keywords:** hepatocytes, intracellular lipid, morphometry, oxidative stress, tissue glycogen

## Abstract

Background/Objectives: Wheat gluten intolerance increases intestinal permeability, triggering inflammation that may directly affect liver function and compromise metabolic health. Methods: Male Wistar rats (n = 50) aged 21 days were divided into five groups (n = 10) based on dietary gluten levels over 100 days: G0 (0%), G14 (14%), G42 (42%), G70 (70%), and G70/0 (70% for the first 70 days, then 0% until euthanasia). At 121 days, the animals were weighed and euthanized, and blood samples were collected for biochemical analyses. Adipose tissue deposits and the liver were excised and weighed. Liver lobes were isolated and fixed for morphological and morphometric analysis of hepatocytes, tissue glycogen percentage, and intracellular lipid assessment. Results: The hepatic oxidative status was evaluated. The ingestion of diets with excess gluten (70%) increased final body mass and reduced liver mass, though it did not alter the adiposity index. Cholesterol, triglycerides, and myeloperoxidase enzyme activity exhibited distinct patterns across all groups. Conclusions: Elevated gluten levels increased oxidative stress and altered tissue hepatic morphology and morphometry.

## 1. Introduction

Gluten is composed of two main protein groups: prolamins and glutenins. Prolamins provide cohesion and extensibility and are classified according to their grain source: gliadin in wheat, hordein in barley, avenin in oats, and secalin in rye [1,2]. Glutenins, present in all these cereals, are responsible for the elasticity of gluten [1].

Tolerance to the amount of ingested gluten varies depending on the organism, as it is directly linked to the presence of specific enzymes capable of breaking down portions of prolamins and glutenins from food. These components are often incompletely digested by gastric, pancreatic, and intestinal peptidases, leading to the formation of large peptides that cross the intestinal epithelial barrier and enter the lamina propria through transcellular or paracellular pathways [3,4]. These peptides can trigger systemic immune responses and contribute to extra-intestinal manifestations, including alterations in liver function. Therefore, gluten intolerance is a broad term encompassing three main types of disorders: celiac disease (CD), wheat allergy, and non-celiac gluten sensitivity (NCGS) [1].

Defined as a set of clinical signs, NCGS is caused by gluten ingestion, leading to intestinal symptoms (such as abdominal pain and tenderness, irregular bowel movements, constipation, or diarrhea) and/or extra-intestinal symptoms, including vomiting, headaches, and liver lesions. These symptoms can be alleviated by removing gluten from the diet. Since it is not an allergic or autoimmune condition, NCGS may be self-diagnosed or diagnosed by healthcare professionals after ruling out celiac disease and wheat allergy through specific tests [4,5,6,7].

In the context of NCGS and considering the inflammatory potential of gluten, studies have shown that it can alter the morphology of the gastrointestinal tract (GIT), which plays a crucial role in nutrient digestion and absorption. Persistent exposure to gluten may increase intestinal permeability and intraepithelial lymphocyte levels, triggering immune responses to dietary and microbial antigens. This cascade can lead to local and systemic inflammation accompanied by oxidative stress—marked by elevated nitric oxide levels—that may reach the liver and contribute to hepatic alterations [5,8,9,10].

The liver, a central component of metabolism, is responsible for processing and storing nutrients and producing plasma proteins such as albumin. These processes are carried out by hepatocytes, the predominant cell population. The liver is involved in detoxifying, neutralizing, and degrading drugs and toxic substances, in addition to playing a key role as a regulator of innate and adaptive immunity, as it is exposed to a large number of foreign molecules originating from the gut [5,11].

The gut–liver axis is characterized by intense blood flow from the intestine to the liver through the hepatic portal vein. It is responsible for transporting substances to be metabolized, especially gluten-derived peptides, which, under specific conditions such as intestinal barrier breakdown, can trigger a hepatic immune response [5]. Dysregulation of the gut–liver axis and activation of immune responses can promote liver damage through inflammatory processes, non-alcoholic fatty liver disease, hepatic fibrosis, and cellular pro-oxidant/antioxidant imbalance. These disruptions may affect cellular proliferation, differentiation, or apoptotic responses and disturb tissue homeostasis [5,9].

Given evidence linking gluten to systemic inflammation and oxidative stress, we evaluated the impact of dietary gluten levels on liver morphophysiology in Wistar rats.

## 2. Materials and Methods

The experimental procedures were approved by the Animal Ethics Committee of the State University of Maringa under protocol number 4375010922 in accordance with the guidelines established by the National Council for the Control of Animal Experimentation.

### 2.1. Experimental Design

#### 2.1.1. Animals

Fifty male Wistar rats (*Rattus norvegicus*) (n = ten per group) were used. The animals were provided by the Central Animal Facility of the State University of Maringa at 21 days of age. The rats were housed for 100 days in the animal facility of the Department of Morphological Sciences at the State University of Maringa under standardized conditions: regular 12 h light/dark cycles, controlled temperature (22 ± 2 °C), and air exhaustion.

The groups were organized according to the wheat gluten levels in the diet:G0: Gluten-free diet;G14: 14% wheat gluten (standard control diet);G42: 42% wheat gluten;G70: 70% wheat gluten;G70/0: 70 days on a high-gluten diet (70%) followed by 30 days on a gluten-free diet.

All animals were euthanized at 121 days of age.

#### 2.1.2. Feed Preparation

The diets were produced at the Department of Food Engineering of the State University of Maringa using a single-screw extruder machine (IMBRA RX50 – INBRAMAQ, Ribeirão Preto, São Paulo, Brazil), which operates at high speed (up to 400 rpm), coupled with an output die and a diffuser piece responsible for shaping the feed into pellets.

All ingredients, in percentages appropriate for rodents, were ground into flour. This product was then moistened and subjected to high temperature (110–160 °C) and pressure to cook, which resulted in pellet expansion. The degree of expansion was controlled by the moisture content (25%). At room temperature, the extruded product was cooled, preventing additional changes.

The feed was dried in an air-circulation oven at 50–60 °C for 24 h to maintain the appropriate hardness level for rodents. The refinement of ingredients in the diet followed the American Institute of Nutrition AIN-93 guidelines [12]. After preparation, the feed was stored under refrigeration in separate containers to prevent gluten cross-contamination. The nutritional values of the standard diet are provided in Table 1.

To determine the gluten levels, wheat flour and rice bran were used as raw materials. In the standard diet of the control group (G14), wheat served as the gluten source (protein component), which was replaced by rice bran in group G0. For groups with gluten levels above 14%, the percentage of wheat was increased. The diet formulation included a vitamin and mineral premix for rodents (PX 1577 PREMIX—Nucleopar Nutrição Animal Ltda., Mandaguari, PR, Brazil). The ingredients selected to produce 1 kg of balanced feed are described in Table 2 and were weighed using an analytical balance (220 g M214Ai Bel^®^, BEL Piracicaba, São Paulo, Brazil) and a Triunfo^®^ digital scale (DST30/C-DM), (Triunfo Indústria de Balanças Eletrônicas Ltda., São Paulo, Brazil).

#### 2.1.3. Food Consumption and Body Mass

To evaluate the animals’ food consumption, the provided feed and leftovers were weighed every two days. The feed allowance was calculated between 30 and 60 g per animal to ensure ad libitum consumption [13]. All animals were weighed weekly on a fixed day to assess weight gain using a Filizola™ digital scale (Indústrias Filizola S.A., São Paulo, Brazil) (model BP3). Changes in body mass were recorded in grams (g). Food consumption and body mass evolution of the animals were expressed in grams (g) and recorded at 1, 25, 50, 75, and 100 days.

**Table 2 nutrients-17-01842-t002:** Components of rodent diets with varying levels of gluten (wheat) labeled as 0% (G0), 14% (G14), 42% (G42), and 70% (G70) [14].

Ingredients (g/kg)	G0	G14	G42	G70
Corn grain	614	614	334	0
Soybean meal	141	141	141	141
Wheat flour	0	140	420	700
Rice bran	140	0	0	0
Soybean oil	40	40	40	40
Sodium chloride	6.8	6.8	6.8	6.8
Calcium carbonate	25	25	25	25
Dicalcium phosphate	11.4	11.4	11.4	11.4
Vitamin premix	10	10	10	10
Mineral premix	11.8	11.8	11.8	11.8

### 2.2. Biological Material Collection

According to the ethical approval code (protocol number 4375010922), at 121 days of age, after overnight fasting, the animals were weighed and euthanized by deepening anesthesia (Ketamine 90 mg/kg body mass + Xylazine 9 mg/kg body mass). The nasoanal length was measured with a metric tape for subsequent calculation of the Lee Index ([body mass (g) × 1000]^1/3^/nasoanal length (cm)) [15]. Blood was collected via cardiac puncture, centrifuged to obtain serum, and stored in a freezer for later analysis.

Following a vertical laparotomy, the liver was collected, weighed, and divided into lobes. Samples from the median lobe were fixed in paraformaldehyde, while other lobes were frozen in liquid nitrogen and stored at −80 °C in a freezer.

Adipose tissue deposits (mesenteric, retroperitoneal, periepididymal, and inguinal) were collected and weighed to determine the adiposity index (sum of adipose tissue deposits/100 g of body mass).

### 2.3. Biochemical Analyses

The collected serum was used for biochemical blood assays, including total protein, albumin, total cholesterol, triglycerides, aspartate aminotransferase (AST), alanine aminotransferase (ALT), and gamma-glutamyl transferase (GGT). Samples were centrifuged at 3000 rpm for 15 min and analyzed using commercial kits (Gold Analisa Diagnostica Ltda., Minas Gerais, Brazil) according to the manufacturer’s specifications in a Bioplus 2000 spectrophotometer (Bioplus Produtos para Laboratórios Ltda., São Paulo, SP, Brazil).

The serum was also used to assess MPO enzyme activity. The samples were homogenized in potassium phosphate buffer (50 mmol L^−1^, pH 6) and centrifuged at 5000 rpm (rotor FA-45-30-11, model 5810R, Eppendorf SE, Hamburg, Germany) at room temperature (25 °C) for 15 min. For the analysis, the pellets resulting from centrifugation were used, which were resuspended in potassium phosphate buffer (50 mmol L^−1^) containing 0.5% hexadecyltrimethylammonium bromide (HTAB, pH 6) and centrifuged at 5000 rpm at 25 °C for 15 min.

To determine MPO activity, the reaction was initiated by diluting o-dianisidine (18.4 mmol L^−1^), and after 5 min at 37 °C, it was stopped by adding sodium acetate (1.46 mol L^−1^, pH 3.0). The readings, performed on a spectrophotometer (FlexStation 3 Multi-Mode Microplate Reader, Molecular Devices LLC, San Jose, CA, USA) at 450 nm, were expressed as optical density (O.D.) units per mg of protein [16].

### 2.4. Processing and Histological and Histochemical Analysis of the Liver

The median liver lobe was fixed in 4% paraformaldehyde for 24 h and subsequently transferred to a 70% alcohol solution. The samples were then subjected to histological processing and paraffin embedding to obtain semi-serial cross-sectional sections (5 μm thick) using a rotary microtome (Leica^®^ RM2245, Leica Biosystems, Nussloch, Germany). Sections were stained with Harris Hematoxylin and Eosin (H&E) and a Periodic Acid–Schiff (PAS) histochemical reaction.

For morphometric analysis of hepatocytes, the area of 100 cells/animal (µm^2^) was measured. All hepatocytes present in 50 images/animal, corresponding to an area of 57,510.05 µm^2^/image, were quantified, and the results are expressed as the number of hepatocytes, adapted from Azevedo et al. [17].

Histochemical analysis to assess glycogen percentage was performed on 50 images/animal (57,510.05 µm^2^/image). All analyses were conducted using Image Pro-Plus^®^ software version 4.5 (Media Cybernetics, Rockville, MD, USA).

The left lateral liver lobe was frozen in liquid nitrogen, stored at −80 °C, and later sectioned into semi-serial cross-sectional histological sections (10 μm thick) using a cryostat (Leica^®^ CM1850) at −24 °C. Sections were subjected to Sudan III histochemical staining to evaluate intracellular lipid inclusion percentages. The area of 100 hepatocytes/animal and their respective lipid inclusions was measured using Image Pro-Plus^®^ software version 4.5 (Media Cybernetics).

All images were captured near the central vein region using an optical microscope (Nikon Eclipse^®^, Shinjuku, Japan) with a 40× objective, coupled to a high-resolution camera (Nikon^®^, DS-Fi1c, Shinjuku, Japan) and connected to a computer running NIS-Elements F software version 4.00.00.

### 2.5. Oxidative State

To comprehensively evaluate hepatic oxidative stress and antioxidant defense mechanisms, specific biochemical markers were selected based on their relevance in detecting protein oxidation, lipid peroxidation, and enzymatic antioxidant activity.

#### 2.5.1. Tissue Preparation

The right posterior, right inferior, and caudate liver lobes were removed, clamped, and frozen in liquid nitrogen. Subsequently, tissue samples were homogenized using a Potter–Elvehjem homogenizer (DWK Life Sciences, Millville, NJ, USA) with ten volumes of ice-cold 0.1 M phosphate buffer (pH 7.4), and an aliquot was separated as the total homogenate. The remaining homogenate was centrifuged (11,000× *g* for 15 min), and the supernatant was separated as the soluble fractions of the homogenate.

#### 2.5.2. Oxidative State Parameters

Carbonylated proteins: Carbonylated protein levels were measured spectrophotometrically using 2,4-dinitrophenylhydrazine (DNPH) at 370 nm = 22 × 10^3^ M^−1^·cm^−1^, with values expressed as nmol/(mg protein) [18].

TBARSs: Thiobarbituric acid reactive substance (TBARS) levels were determined spectrophotometrically (502 nm), as described by Buege and Aust [19]. Standard curves were prepared using 1,1,3,3-tetraethoxypropane, and the results are expressed as nmol/(mg protein).

Catalase: Catalase activity was estimated by monitoring absorbance changes at 240 nm using H_2_O_2_ as the substrate, with activity expressed as mmol/(min × mg protein) [20].

Superoxide dismutase (SOD): SOD activity was assayed based on its ability to inhibit pyrogallol autoxidation in an alkaline medium, measured at 420 nm [21]. One unit of SOD was defined as the amount of enzyme required to inhibit pyrogallol oxidation by 50%, with results expressed as units/(mg protein). Total protein was quantified in the homogenate and supernatant using the Folin phenol reagent [22].

### 2.6. Statistical Analysis

The data were analyzed using GraphPad Prism software (GraphPad Software, San Diego, CA, USA), version 8.0. For all analyses, QQ-plot was used to visually inspect for deviations from normality, and one-way ANOVA was applied, followed by Tukey’s post-hoc test, with results expressed as mean ± standard deviation. In all cases, a *p*-value < 0.05 was considered statistically significant.

## 3. Results

### 3.1. Food Consumption and Body Mass Progression

Figure 1a shows the evolution of feed consumption (g) per group during the 100-day experimental period. Qualitatively, the group receiving 0% gluten exhibited the highest food intake, followed by the group fed a 70% wheat gluten diet for 70 days and then switched to a 0% gluten diet for the final 30 days. The graph indicates that the latter group experienced a potential adaptation period following the diet change on day 70, characterized by a decline in consumption, followed by a restoration of the previously observed parameters.

At the beginning of the experiment, the group receiving 14% gluten exhibited the lowest feed consumption rate, which persisted until the final days when it matched group G0. The values recorded for the groups fed 42% and 70% gluten remained intermediate between those of the G0 and G70/0 groups (which showed higher feed consumption) and the G14 group (which showed lower feed consumption). However, by the end of the experiment, groups G42 and G70 had the lowest average feed consumption.

Figure 1b presents the weekly body mass gain among the experimental groups. Groups G70 and G70/0 exhibited the highest exponential increase in body mass. However, in the final weeks of the experiment, the group that received a 70% gluten diet for the first 70 days and a 0% gluten diet for the last 30 days showed stabilization in body mass gain, likely due to the change in diet composition.

The animals in group G42 displayed body mass gains throughout the experiment, similar to those fed a high-gluten diet. Conversely, the animals fed 0% and 14% gluten diets showed lower mass gains, which were comparable between these two groups.

### 3.2. Biometric Parameters

Table 3 presents the data obtained on the final day of the experiment for assessing the biometric parameters of fasted rats. The final body mass shows a significant increase in groups G70 and G70/0 compared to group G14, complementing the findings illustrated in Figure 1. Similar results were observed in the nasoanal length values, with a significant increase in groups G70 and G70/0 compared to G14. However, there was also a significant decrease in G14 compared to G0. Liver mass was significantly lower in the groups fed a high-gluten diet (G70 and G70/0) compared to G0. The Lee Index did not show significant differences between groups.

### 3.3. Adipose Tissue

Although a significant difference in final body masses was observed, only the periepididymal adipose tissue deposits in groups G70 and G70/0 showed a significant difference compared to the control group (G14). Regardless of the experimental group, there were no statistical differences in mesenteric, retroperitoneal, or inguinal adipose tissue mass or in the adiposity index (Table 4).

### 3.4. Biochemical Parameters

The results of the blood biochemical analyses are described in Table 5. Total protein (TP) and albumin (ALB) levels remained constant across all groups, indicating that, regardless of the gluten level in the diet, all diets were characterized as normoproteic.

Total cholesterol (TC) showed a significant decrease in groups G14 and G70/0 compared to the group fed only a 0% gluten diet. In triglyceride (TG) analysis, a significant increase was observed in group G42 compared to the control group. Liver injury markers—transaminases (AST and ALT) and gamma-glutamyl transferase (GGT)—did not differ significantly between groups, indicating no detectable tissue damage based on these markers.

Myeloperoxidase (MPO) activity was significantly increased in the G0 group compared to G14, G42, and G70/0. The G70 group also showed higher MPO activity than the control group (G14) (Table 5).

### 3.5. Morphological and Morphometric Analyses

Macroscopically, the livers of animals fed diets with high wheat gluten levels (G70 and G70/0) exhibited striations and whitish spots on the surface, differing from the other groups. Microscopically, they retained standard histological organization, with no signs of fibrosis, inflammatory aggregates, or tissue necrosis (Figure 2b,d). However, lipid deposition (vacuolization) was pronounced in the groups with excess or high gluten intake (G42, G70, and G70/0).

Hepatocyte counts were obtained from a standardized area of 57,510.05 µm^2^ across 50 images per animal, captured near the central vein. A significant increase in hepatocyte number was observed in all gluten-fed groups (G14, G42, G70, and G70/0) (*p* < 0.05) compared to G0, while a reduction in average hepatocyte count was seen in G42, G70, and G70/0 compared to the control group (G14). The group fed 70% wheat gluten until 91 days of age and a gluten-free diet until euthanasia (G70/0) showed a significant increase in hepatocyte number compared to G42 and G70, which exhibited no statistical differences between them (Figure 2a).

In contrast to the results for hepatocyte numbers, the hepatocyte cell area showed a significant reduction in all gluten-fed groups (G14, G42, G70, and G70/0) compared to G0. It increased in group G42 compared to G14 and decreased in groups G70 and G70/0 compared to G42 (Figure 2c). Thus, increasing dietary gluten levels promoted a rise in the hepatocyte number and a reduction in cell area (profile).

The tissue glycogen percentage and intracellular lipid percentage across groups are shown in Figure 2e and Figure 2g, respectively. The glycogen results were directly proportional to the liver cell area: a significant reduction was observed in all gluten-fed groups (G14, G42, G70, and G70/0) compared to G0; an increase in the 42% gluten group compared to the control (G14); and a decrease in G70 and G70/0 compared to G42. Figure 2f demonstrates the presence of tissue glycogen through the PAS histochemical reaction.

Lipid inclusions in hepatocytes were assessed using the Sudan III histochemical reaction on frozen sections (Figure 2h). Higher dietary gluten concentrations correlated with increased lipid accumulation in hepatocytes. Significant increases were observed in groups G14, G42, G70, and G70/0 compared to G0; in groups G42, G70, and G70/0 compared to the control group (G14); in G70 and G70/0 compared to the 42% gluten-fed group; and in G70 compared to G70/0. Qualitatively, the highest lipid deposition was observed in the livers of the 42% and 70% gluten diet groups compared to G0 and G14 (Figure 2h).

### 3.6. Oxidative State

The hepatic tissue oxidative stress markers—carbonylated proteins and TBARSs—across experimental groups are shown in Figure 3a and Figure 3b, respectively. For carbonylated proteins (Figure 3a), a significant increase was observed in all gluten-fed groups (G14, G42, G70, and G70/0) compared to the gluten-free group (G0). The diet with the highest gluten level (G70) exhibited the greatest concentration of carbonylated proteins, differing significantly even from the control group (G14) and G0. Removing gluten from the diet for 30 days in group G70/0 led to a significant reduction in this marker compared to groups G42 and G70, suggesting a positive effect. Figure 3b, displaying the TBARS results, shows a significant increase only in groups G14, G42, and G70 compared to G0. Overall, gluten intake was associated with increased oxidative stress in hepatic tissue, while gluten withdrawal appeared to partially reverse these effects.

Figure 3c,d present the hepatic antioxidant activity of catalase and superoxide dismutase (SOD) across groups. The catalase assay reveals a significant decrease in groups G42 and G70 compared to G0, suggesting that gluten intake may impair antioxidant defense in the liver, leading to reduced catalase activity. However, group G70/0 showed a significant increase compared to G42 and G70, indicating that removing gluten from the diet may help restore antioxidant capacity and potentially alleviate oxidative stress in the liver. The SOD activity results indicate a significant decrease only in groups G42 and G70 compared to G0, further supporting the idea that gluten consumption contributes to oxidative damage in hepatic tissue. These findings suggest that gluten-induced changes in antioxidant enzyme activity could contribute to liver dysfunction and that gluten withdrawal may have therapeutic potential in restoring hepatic antioxidant defense and reducing oxidative stress.

## 4. Discussion

Non-celiac gluten sensitivity (NCGS) is defined by intestinal and extra-intestinal symptoms triggered by gluten ingestion in individuals without celiac disease or wheat allergy [23]. As a result, interest in gluten-free diets has grown rapidly, fueled by media attention and medical discussions linking gluten to symptoms like diarrhea, vomiting, and constipation in humans [23,24].

Due to this, researchers warn about potential nutritional concerns regarding the complete removal of gluten from the diet, including deficiencies in nutrients such as B-complex vitamins, minerals, and fiber. Compensatory increases in the consumption of saturated and hydrogenated fatty acids and high carbohydrate content may lead to a rise in the glycemic index in humans [25] and rats [26].

However, cereals such as wheat remain widely consumed worldwide, particularly in the Mediterranean diet, which is regarded as the healthiest dietary pattern in existence [27]. Considering that wheat gluten composition is estimated to be approximately 85% carbohydrates, 9–18% proteins, and 3–4% lipids [28], the standardized feed produced for rodents for this experiment contained 14% wheat gluten.

Table 1, through calorimetric analysis, presents the following nutritional values for the standard diet: 75.4% carbohydrates, 14.1% proteins, and 4% lipids. Thus, the feed used is normoproteinic, as it did not exceed levels greater than 80% carbohydrates or 19% proteins [29], and is effective in maintaining the animals’ body mass gain. Since it contains more than 1% lipids, it should be considered hypercaloric [29].

In a feeding process, the metabolizable energy requirement must be appropriate, as the efficiency of ingested energy (gross energy) is utilized to regulate the animals’ vital functions (net energy). In other words, energy demands directly influence the quantity of food consumed [30]. Animals fed the standard diet (14%) exhibited the lowest food consumption qualitatively, demonstrating that the feed can adequately supply the nutrients required for development and maintenance (Table 1 and Table 2). Conversely, the groups with the highest food consumption were the gluten-free group (G0), followed by the group fed a 70% wheat gluten diet for 70 days and then a gluten-free diet until euthanasia (G70/0), possibly due to changes in diet composition affecting energy demands.

An epidemiological study by Bektaş et al. [27] reported that 65% of Americans believe gluten-free diets are healthier, and 27% assume eliminating gluten leads to weight loss. However, the literature lacks sufficient evidence linking gluten intake to obesity development in mice [31]. Our results align with findings in mice, as no difference in body mass was observed between rats fed a completely gluten-free diet (G0) and those receiving standard (14%) or excessive (42%) gluten levels in rodent diets, despite higher food consumption in the G0 group.

However, the group of animals fed a 70% gluten diet—i.e., a gluten overload—showed a distinct response compared to the standard diet (14%), with a significant increase in final body mass, nasoanal length, food consumption, and Lee Index (*p* > 0.05). Contrasting results were reported by Bektaş et al. [27], where Wistar rats fed a standard diet supplemented with 12% wheat gluten showed no differences in final body mass or food intake.

The G70/0 group was designed to assess the animals’ response to a 70-day gluten overload, followed by a 30-day gluten-free diet. Behavior similar to G70 was observed in the aforementioned parameters, likely due to the short withdrawal period of gluten restriction, preventing reversal of the condition.

Liver mass remained stable in association with gluten levels in the diet (0%, 14%, or 42%) (Table 3). Similar results were reported by Liang et al. [32] in golden hamsters fed a gluten-containing (standard) diet and a gluten-free diet for five weeks. However, gluten overload (G70 and G70/0) caused a significant reduction in liver mass compared to the G0 group.

Of the four adipose tissue deposits evaluated (mesenteric, retroperitoneal, periepididymal, and inguinal), only the periepididymal deposit showed an increase in groups G70 and G70/0 compared to the control group (G14). Despite this, the adiposity index remained stable. This response may be associated with the maintenance of the final body mass in the gluten-free, control, and high-gluten groups. When assessing visceral fat, Freire et al. [31] reported similar findings, observing no significant differences in non-obese mice fed a standard rodent diet supplemented with 4.5% gluten for eight weeks.

The analysis of the lipid profile indicates that the percentage of gluten in the diet influences total cholesterol and triglyceride levels. Total cholesterol plays a significant role in diagnosing various diseases, including hepatic disorders [33]. Animals fed a gluten-free diet exhibited the highest cholesterol levels among all groups. We attribute this to the inclusion of rice bran, which replaced gluten, as it contains a high carbohydrate content (up to 30% starch) and approximately 15% fiber [34]. Rice bran is produced through the separation of the husk and the grain, primarily composed of lignin and silica, resulting in very low nutritional value. Additionally, rice bran can undergo lipid peroxidation and may contain anti-nutritional factors that interfere with lipid metabolism [34].

The groups receiving higher gluten levels (G42, G70, and G70/0) also exhibited significantly elevated triglyceride levels, particularly between G42 and G14. All ingested food that is not immediately utilized is converted into triglycerides and stored in adipocytes, which can later be mobilized as an energy source via hormonal regulation [33]. Notably, the total cholesterol and triglyceride values for all groups did not fall within the reference ranges for adult Wistar rats [35], indicating that the diet provided negatively impacted the lipid profile. However, this did not translate into changes in the adiposity index. It is also worth noting that the deviation from reference ranges might reflect methodological limitations or biological variability rather than pathological outcomes alone, and these factors should be considered when interpreting the lipid profile results.

Gluten levels had no influence on blood parameters, including total protein (TP), albumin, liver transaminases (aspartate aminotransferase (AST) and alanine aminotransferase (ALT)), and gamma-glutamyl transferase (GGT). For TP and albumin, the results align with the components of the normoproteic diets produced (Table 1).

AST and ALT are enzymes predominantly found in the liver and are valuable for diagnosing hepatic disorders [36]. Elevated AST levels are strongly associated with hepatic fibrosis [37]. The results for AST were not significant (*p* > 0.05); however, there was a notable trend toward higher levels in gluten-fed groups compared to the gluten-free group. ALT levels also showed no significant differences between groups, though the control group exhibited higher values compared to G0, G42, and G70. Importantly, despite quantitative variations, both AST and ALT levels remained within reference ranges for Wistar rats [35].

Similar to transaminases, GGT (gamma-glutamyl transferase) is a hepatic enzyme also found in renal tubules, the pancreas, and the intestines. Elevated serum GGT levels are associated with liver diseases of any etiology, particularly linked to fibrosis [38]. Serum GGT levels showed no differences between experimental groups, indicating an absence of hepatic tissue injury. Similar results were observed by Ozuna and Barro [26], who compared increasing levels of gliadin (the protein component of gluten) in rodent diets and found no significant differences in AST, ALT, or GGT levels.

Although the AST, ALT, and GGT levels did not differ significantly among the experimental groups, these enzymes remain essential markers for monitoring liver health. Subtle, non-significant changes may reflect early or subclinical alterations in hepatic function, especially in experimental contexts involving dietary interventions. The trend toward elevated AST in gluten-fed groups and the variation in ALT levels suggest that gluten intake may influence liver physiology, even in the absence of overt damage [39].

Immune responses are initiated through the recruitment and activation of innate immune cells, including neutrophils [40]. These cells contain potent enzymes such as myeloperoxidase (MPO), which plays a crucial role in the host defense system by generating reactive oxidants to neutralize pathogens [40,41]. However, excessive production of these oxidants can lead to tissue damage and amplify inflammatory responses [41].

In line with this, Chen et al. [42] reported increased circulating MPO levels in rats following a three-week intravenous administration of emulsified gliadin compared to untreated controls. Similarly, in the present study, MPO activity was significantly elevated in the G70 group compared to the control group (G14), indicating a strong pro-inflammatory effect of high dietary gluten exposure. This difference was statistically significant (*p* < 0.05), reinforcing the robustness of the association between high gluten intake and neutrophil-mediated inflammatory responses.

There was a significant increase in myeloperoxidase (MPO) levels in the gluten-free group (G0) compared to groups G14, G42, and G70/0. It is hypothesized that this alteration may also be linked to the inclusion of rice bran in the diet composition, which replaced wheat gluten, as discussed earlier. Components in portal blood, such as gut-derived elements (including gluten-derived peptides) and bacterial products, can, under specific conditions, trigger immune responses associated with hepatic inflammation and fibrosis [5].

The livers of animals fed a gluten overload (70% and 70/0%) exhibited localized whitish spots and striations. However, elevated wheat gluten levels did not disrupt the standard hepatic histological organization. Microscopically, hepatocyte cords, with central nuclei interspersed by sinusoids converging toward the central vein (Figure 2b), were preserved. No inflammatory cell infiltrates were observed, though intracellular lipid deposition was increased in groups G42, G70, and G70/0.

We observed a relationship between gluten-containing feed consumption and liver tissue morphometry. Analysis of the median liver lobe in animals fed gluten, regardless of the level, reveals a significant increase in hepatocyte number and a reduction in cellular profile (area) compared to the gluten-free group. This effect aligns with liver mass data: gluten-containing diets resulted in smaller livers with more numerous but smaller hepatocytes. This response may represent a compensatory mechanism, where hepatocyte proliferation rates increase due to an imbalance in dietary peptide metabolism, acting as a stressor. Similar morphometric adaptation responses (cell number and profile) in the hepatocytes of rats receiving dietary supplementation correlated with organ weight were also reported by Azevedo et al. [17].

Hepatic glycogen and lipid deposits were assessed using PAS and Sudan III histochemical staining. The results for glycogen showed a lower percentage in the liver tissue of animals fed a gluten-containing diet. Glycogen levels are influenced by the animal’s nutritional status. This status determines which metabolic pathways are activated for energy production [43]. Therefore, the lower glycogen content may indicate increased glycogenolysis, where glycogen is broken down into glucose to meet energy demands.

There was a significant increase in lipid inclusions in the hepatocytes of animals fed gluten compared to the gluten-free group (G0). This suggests a direct relationship between dietary gluten intake and lipid storage in hepatocytes. The dose-dependent response further supports this association. Significant increases in lipid deposits were observed in groups receiving excess (42%) and an overload (70%) of wheat gluten compared to the control group (14%). Similar findings were reported in mice fed gluten, where hepatocytes contained large lipid droplets indicative of hepatic steatosis [44].

The increasing addition of gluten to the diet induces elevated oxidative stress in the livers of rats. A significant increase in carbonylated protein (pro-oxidant) levels was observed in gluten-fed groups compared to the gluten-free group (G0). Compared to the gluten control group (14%), the group receiving a gluten overload (70%) showed a significant rise in oxidative stress. The G70/0 group exhibited significantly lower levels than the excess (42%) and overloaded (70%) gluten groups, approaching equivalence with the G0 levels.

Regarding TBARSs (pro-oxidant), there was an increase in levels for groups G14, G42, and G70 compared to the gluten-free group (G0). The control group (G14) did not differ from the others and maintained levels similar to the group fed 42% wheat gluten. The G70/0 group showed a reduction compared to the group fed only 70% gluten, though this difference was not significant, aligning with the results for carbonylated proteins and trending toward equivalence with the G0 levels. Elevated TBARSs and the shift in the pro-oxidant/antioxidant cellular balance may affect cell proliferation, differentiation, or apoptotic responses, disrupting tissue homeostasis [9]. Therefore, we propose that TBARSs may have stimulated cellular proliferation, contributing to the increased number of hepatocytes in the gluten-fed groups.

As expected, antioxidant enzyme activity (catalase and SOD) was reduced in animals fed diets containing gluten. Catalase activity showed a significant decrease in groups G42 and G70 compared to the gluten-free group (G0), indicating a direct relationship between increased dietary gluten and diminished antioxidant activity. The G70/0 group exhibited the highest catalase activity, which was significantly greater than in groups receiving excess (42%) or an overload (70%) of gluten.

The G0 group exhibited higher SOD activity compared to other groups, though this was significant only relative to groups G42 and G70. The control group showed no differences from the remaining groups. The group fed a 70% wheat gluten diet for 70 days, followed by a gluten-free diet until euthanasia (G70/0), displayed higher SOD levels compared to the group fed only 70% gluten (G70), though no significant differences were observed.

The differential behavior between catalase and SOD may reflect distinct regulatory mechanisms and cellular roles for each enzyme. Catalase primarily detoxifies hydrogen peroxide within peroxisomes, whereas SOD dismutates superoxide radicals in both the cytosol and mitochondria. These compartmental and functional differences might result in varying sensitivities to gluten-induced oxidative stress [45].

There is a clear interference of dietary wheat gluten levels on tissue oxidative status. According to Aguilar et al. [46], wheat gluten consumption induces hepatic steatosis in obese mice, exacerbating inflammation, reactive oxygen species (ROS) production, and lipid peroxidation while reducing the activity of SOD and catalase enzymes. We propose that when gluten was removed from the diet (G70/0), oxidative stress in the liver also decreased, as evidenced by reduced TBARS and carbonylated protein levels, along with increased activity of the antioxidant enzyme catalase. In this context, removing gluten from the diet could positively reduce oxidative stress in hepatic tissue.

Based on the experimental design conducted, the use of diets with gluten-free and increasing gluten levels did not elevate the adiposity index, and no evidence of hepatic injury was detected. However, effects were observed on periepididymal adipose tissue, body mass, liver mass, inflammation levels, hepatic morphophysiology, and oxidative status. These findings align with and expand upon previous clinical observations indicating a link between gluten-related disorders and liver alterations, such as the increased prevalence of nonalcoholic fatty liver disease (NAFLD) in celiac disease (CD) patients [47], as well as the immunological involvement suggested by anti-*Saccharomyces cerevisiae* antibody (ASCA) positivity in CD [48].

Although our study was conducted in an experimental setting, the results highlight the potential metabolic and hepatic impacts of gluten exposure, reinforcing the importance of dietary management in individuals with gluten sensitivity or CD. Furthermore, these findings open new avenues for investigating tailored nutritional strategies and anti-inflammatory approaches aimed at modulating the gut–liver axis, which may have therapeutic relevance in preventing or decreasing liver-related comorbidities in gluten-related disorders.

## 5. Conclusions

The NCGS model used here demonstrates that high dietary gluten induces significant changes in liver morphology and oxidative stress markers, indicating potential hepatic implications, even in the absence of overt injury.

## Figures and Tables

**Figure 1 nutrients-17-01842-f001:**
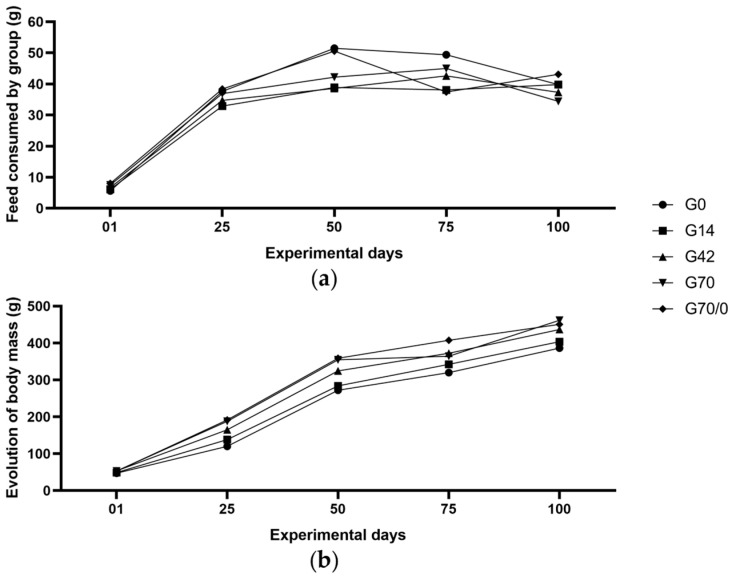
Food consumption and body mass progression (g): (**a**) trend in feed consumption; (**b**) weekly body mass progression (expressed as age in days per group) during the experimental period in rats fed diets with varying levels of gluten (wheat), labeled as 0% (G0), 14% (G14), 42% (G42), 70% (G70), and 70% for 70 days, followed by 0% for 30 days (G70/0). Results are expressed as mean (n = 9–10 per group).

**Figure 2 nutrients-17-01842-f002:**
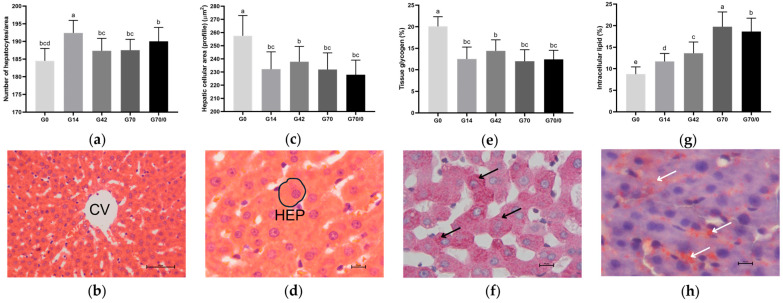
Morphological analyses and liver photomicrographs of rats fed diets with varying levels of gluten (wheat), labeled as 0% (G0), 14% (G14), 42% (G42), 70% (G70), and 70% for 70 days, followed by 0% for 30 days (G70/0): (**a**) number of hepatocytes in a standardized area of 57,510.05 µm^2^/50 images per animal; (**b**) hepatocyte cords and sinusoids converging toward the central vein (CV), H&E staining, 40× objective, scale bar = 50 µm; (**c**) hepatocyte cell area measured in 100 hepatocytes/animal, expressed in µm^2^; (**d**) hepatocyte area demarcation (HEP), H&E staining, 100× objective, scale bar = 10 µm; (**e**) hepatic glycogen content (%); (**f**) PAS histochemical reaction highlighting tissue glycogen (black arrow), 100× objective, scale bar = 10 µm; (**g**) intracellular lipid content in 100 hepatocytes/animal (%); (**h**) Sudan III histochemical reaction for intracellular lipid visualization (white arrow), 100× objective, scale bar = 10 µm. One-way ANOVA followed by Tukey’s post hoc test, expressed as mean ± standard deviation (n = 8–10 per group). Means followed by different superscript letters are significantly different at the 0.05 level.

**Figure 3 nutrients-17-01842-f003:**
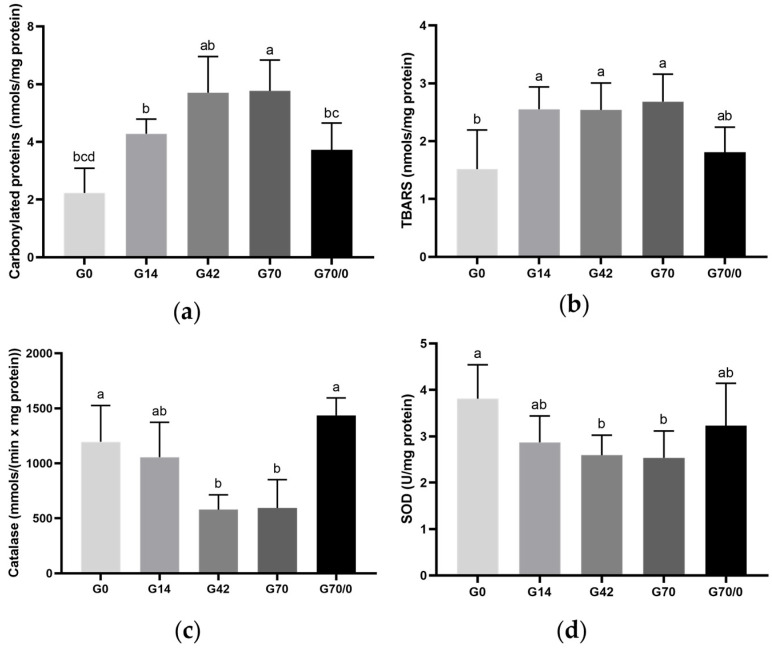
Oxidative state of the liver in rats fed diets with varying levels of wheat gluten, labeled as 0% (G0), 14% (G14), 42% (G42), 70% (G70), and 70% for 70 days, followed by 0% for 30 days (G70/0): (**a**) carbonylated protein expressed as nmol/(mg protein); (**b**) thiobarbituric acid reactive substances (TBARSs) expressed as nmol/(mg protein); (**c**) catalase activity expressed as mmol/(min × mg protein); (**d**) superoxide dismutase (SOD) activity expressed as U/mg protein. One-way ANOVA followed by Tukey’s post hoc test, expressed as mean ± standard deviation (n = 5–7 per group). Means followed by different superscript letters are significantly different at the 0.05 level.

**Table 1 nutrients-17-01842-t001:** Chemical composition and caloric content of the diet with 14% wheat gluten (standard diet) for rodents.

Nutritional Values	Standard Diet
(Kcal per 100 g of feed) *	417
Carbohydrates (%)	75.4
Proteins (%)	14.1
Fats (%)	4
Water (%)	25

Gross energy composition (Kcal per 100 g of feed) *, Laboratory of Food and Animal Nutrition Analysis, Department of Animal Science, State University of Maringa.

**Table 3 nutrients-17-01842-t003:** Biometric parameters: final body mass (FBM) (g), nasoanal length (NAL) (cm), liver mass per 100 g of body mass (LM/100 g), and Lee Index (LI) (g/cm^3^) of rats fed diets with varying levels of gluten (wheat), labeled as 0% (G0), 14% (G14), 42% (G42), 70% (G70), and 70% for 70 days, followed by 0% for 30 days (G70/0).

	G0	G14	G42	G70	G70/0
FBM	436.78 ± 23.75 ^ab^	403.50 ± 29.84 ^b^	436.90 ± 52.23 ^ab^	461.78 ± 39.73 ^a^	450.50 ± 21.62 ^a^
NAL	24.50 ± 0.35 ^a^	23.60 ± 0.84 ^b^	24.00 ± 0.58 ^ab^	24.39 ± 0.42 ^a^	24.40 ± 0.52 ^a^
LM/100 g	3.78 ± 0.29 ^a^	3.58 ± 0.18 ^ab^	3.54 ± 0.32 ^ab^	3.41 ± 0.21 ^b^	3.38 ± 0.15 ^b^
LI	3.09 ± 0.05	3.13 ± 0.06	3.16 ± 0.10	3.17 ± 0.09	3.14 ± 0.06

One-way ANOVA followed by Tukey’s post hoc test, expressed as mean ± standard deviation (n = 9–10 per group). Means followed by different superscript letters are significantly different at the 0.05 level.

**Table 4 nutrients-17-01842-t004:** Adipose tissue (g/100 g of body mass): mesenteric (ME), retroperitoneal (RP), periepididymal (PE), inguinal (IG), and adiposity index (AI) of rats fed diets with varying levels of gluten (wheat), labeled as 0% (G0), 14% (G14), 42% (G42), 70% (G70), and 70% for 70 days, followed by 0% for 30 days (G70/0).

	G0	G14	G42	G70	G70/0
ME	0.98 ± 0.24	0.97 ± 0.27	0.89 ± 0.25	1.14 ± 0.33	1.17 ± 0.28
RP	3.02 ± 0.30	3.11 ± 0.32	3.00 ± 0.28	3.05 ± 0.29	3.27 ± 0.49
PE	2.42 ± 0.35 ^ab^	2.02 ± 0.55 ^b^	2.35 ± 0.37 ^ab^	2.64 ± 0.49 ^a^	2.79 ± 0.51 ^a^
IG	3.06 ± 0.29	2.52 ± 0.26	2.58 ± 0.62	2.69 ± 0.75	2.72 ± 0.44
AI	9.48 ± 0.76	8.62 ± 1.12	8.82 ± 1.20	9.52 ± 1.71	9.96 ± 1.53

One-way ANOVA followed by Tukey’s post hoc test, expressed as mean ± standard deviation (n = 9–10 per group). Means followed by different superscript letters are significantly different at the 0.05 level.

**Table 5 nutrients-17-01842-t005:** Blood biochemical parameters of rats fed diets with varying levels of wheat gluten, labeled as 0% (G0), 14% (G14), 42% (G42), 70% (G70), and 70% for 70 days, followed by 0% for 30 days (G70/0): total protein (TP) (g/dL), albumin (ALB) (g/dL), total cholesterol (TC) (mg/dL), triglycerides (TG) (mg/dL), aspartate aminotransferase (AST) (U/L), alanine aminotransferase (ALT) (U/L), Gamma-glutamyl transferase (GGT) (U/L), myeloperoxidase enzyme (MPO) (OD/mg protein).

	G0	G14	G42	G70	G70/0
TP	5.68 ± 0.42	5.59 ± 0.34	5.63 ± 0.24	5.82 ± 0.92	5.65 ± 0.23
ALB	2.62 ± 0.21	2.43 ± 0.22	2.49 ± 0.23	2.41 ± 0.19	2.54 ± 0.22
TC	140 ± 38.77 ^a^	107.40 ± 11.37 ^b^	119.5 ± 34.40 ^ab^	115.3 ± 24.33 ^ab^	100.70 ± 14.33 ^b^
TG	209.50 ± 50.17 ^ab^	179.29 ± 68.31 ^b^	268.71 ± 81.27 ^a^	244.50 ± 102.92 ^ab^	241.50 ± 72.07 ^ab^
AST	81.29 ± 11.80	96 ± 22.63	93.86 ± 11.98	88.29 ± 35.90	78.57 ±11.44
ALT	45 ± 13.56	61.29 ± 13.03	54.57 ± 15.06	48.71 ± 16.06	61 ± 17.74
GGT	2.14 ± 1.46	2.14 ± 1.46	1.14 ± 1.07	2.29 ± 2.06	1.43 ± 0.98
MPO	0.21 ± 0.04 ^a^	0.07 ± 0.02 ^bc^	0.09 ± 0.01 ^b^	0.15 ± 0.08 ^ab^	0.12 ± 0.03 ^b^

One-way ANOVA followed by Tukey’s post hoc test, expressed as mean ± standard deviation (n = 9–10 per group). Means followed by different superscript letters are significantly different at the 0.05 level.

## Data Availability

The data presented in this study are available upon request from the corresponding author due to ethical reasons.

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
