# Peer review of "Non-Celiac Wheat Gluten Sensitivity Model: Effects on Hepatic Morphophysiology of Wistar Rats"

_nutrients, 2025, doi:10.3390/nu17111842_

Round 1

Reviewer 1 Report

Comments and Suggestions for Authors

In this study, the Authors aimed to evaluate the effects of diets with increasing levels of wheat gluten on the liver morphophysiology of Wistar rats. Fifty male Wistar rats (n=10 per group) were used and split according to the wheat gluten levels in the diet: G0 (Gluten-free diet), G14: 14% wheat gluten (standard control diet), G42: 42% wheat gluten; G70: 70% wheat gluten, G70/0: 70% wheat gluten until 91 days of age (70 days of experimental feeding) followed by a gluten-free diet until euthanasia. All animals were euthanized at 121 days of age. Blood samples were collected for biochemical analyses. Adipose tissue deposits and the liver were excised and weighed. Liver lobes were isolated and fixed for morphological and morphometric analysis of hepatocytes, tissue glycogen percentage, and intracellular lipid assessment. Hepatic oxidative status was also evaluated. The diets were produced at the Department of Food Engineering, and all ingredients, in percentages appropriate for rodents, were ground into flour.

The study showed that the total protein and albumin levels remained constant across all groups, indicating that, regardless of the gluten level in the diet, all diets were characterized as normoproteic. Total cholesterol significantly decreased in groups G14 and G70/0 compared to the group fed only a 0% gluten diet. In triglyceride analysis, a significant increase was observed in group G42 compared to the control group. Liver injury markers—AST, ALT, and gamma-glutamyl transferase—did not differ significantly between groups, indicating no detectable tissue damage based on these markers. Myeloperoxidase enzyme activity results revealed a significant increase in group G0 compared to groups G14, G42, and G70/0. Group G70 showed a significant increase compared to the control group (G14). Thus, the authors concluded that the non-celiac gluten sensitivity (NCGS) model negatively impacts hepatic morphophysiology, increasing inflammation and oxidative stress. Higher dietary gluten levels  (70%) result in smaller livers with reduced hepatocyte area, increased hepatocyte count, and elevated intracellular lipid content and disrupted hepatic morphology and morphometry.

The study is of interest since it might add novel evidence on the importance of gut-liver axis alteration in the pathogenesis of hepatic function and morphology, and activation of immune responses that can promote liver damage through inflammatory processes, non-alcoholic fatty liver disease, hepatic fibrosis, and cellular prooxidant/antioxidant imbalance. 

However, to enhance the clinical impact of this interesting animal pathogenetic model of study in common human diseases, the authors should recall and discuss the literature data on how intestinal permeability is supported as a mechanism that causes both hepatic steatosis and immune dysregulation. In this regard, celiac disease (CD) represents a disease model characterized by both events, as evidenced by the presence of a significant rate of CD patients who develop hepatic steatosis even after the introduction of a gluten-free diet, as previously shown (doi: 10.1111/apt.14910 ), and by the development of an immune response against microbial antigens, as found in a high rate of celiac patients who develop IgA and IgG antibodies against Saccharomyces cerevisiae (ASCA) before starting the gluten-free diet as a consequence of the altered intestinal permeability that characterizes untreated celiac disease, as previously demonstrated (doi: 10.1111/j.1365-2036.2005.02417.x.). These literature studies could support the clinical relevance of the study results and open new therapeutic perspectives in gluten-related disorders.

Author Response

However, to enhance the clinical impact of this interesting animal pathogenetic model of study in common human diseases, the authors should recall and discuss the literature data on how intestinal permeability is supported as a mechanism that causes both hepatic steatosis and immune dysregulation. In this regard, celiac disease (CD) represents a disease model characterized by both events, as evidenced by the presence of a significant rate of CD patients who develop hepatic steatosis even after the introduction of a gluten-free diet, as previously shown (doi: 10.1111/apt.14910 ), and by the development of an immune response against microbial antigens, as found in a high rate of celiac patients who develop IgA and IgG antibodies against Saccharomyces cerevisiae (ASCA) before starting the gluten-free diet as a consequence of the altered intestinal permeability that characterizes untreated celiac disease, as previously demonstrated (doi: 10.1111/j.1365-2036.2005.02417.x.). These literature studies could support the clinical relevance of the study results and open new therapeutic perspectives in gluten-related disorders.

Answer:

Dear Reviewer,

On behalf of our team, I would like to thank you for your thoughtful comments. Discussing the clinical impact of this animal model on common human diseases was invaluable. Thanks to your input, we were able to strengthen our discussion and clearly convey our intention that, through further exploratory experiments, we will achieve results that improve human quality of life. Thus, we add the following paragraphs to the end of the discussion:

“Based on the experimental design conducted, the use of diets with gluten-free and increasing gluten levels did not elevate the adiposity index, and no evidence of hepatic injury was detected. However, effects were observed on periepididymal adipose tissue, body mass, liver mass, inflammation levels, hepatic morphophysiology, and oxidative status. These findings align with and expand upon previous clinical observations indi-cating a link between gluten-related disorders and liver alterations, such as the in-creased prevalence of nonalcoholic fatty liver disease (NAFLD) in celiac disease (CD) patients [47], as well as the immunological involvement suggested by anti-Saccharomyces cerevisiae antibodies (ASCA) positivity in CD [48].

Although our study was conducted in an experimental setting, the results highlight the potential metabolic and hepatic impacts of gluten exposure, reinforcing the im-portance of dietary management in individuals with gluten sensitivity or CD. Fur-thermore, these findings open new avenues for investigating tailored nutritional strategies and anti-inflammatory approaches aimed at modulating the gut-liver axis, which may have therapeutic relevance in preventing or decreasing liver-related comorbidities in gluten-related disorders.”

Reviewer 2 Report

Comments and Suggestions for Authors

Title:

Non-celiac wheat gluten sensitivity model: effects on hepatic 2 morphophysiology of Wistar rats 3

# General:

–Positive Aspects:

  • The study employs a controlled animal model to address non-celiac gluten sensitivity.
  • The experimental design is presented in a clear and coherent manner, with a comprehensive investigation into morphological, biochemical, and oxidative parameters in liver tissue. This is highly relevant for understanding gluten-related disorders.
  • The integration of morphometric, histochemical, and biochemical analyses provides a multidimensional view of the pathological changes induced by varying dietary gluten levels.
  • The manuscript is well organised into standard sections (Introduction, Materials and Methods, Results, Discussion, Conclusions).
  • The methods section is detailed, with specifications of animal handling, feed preparation, and biochemical assays, which supports replication of the study.
  • The use of tables and figures to present the data enhances the visualisation of trends.

–Areas for improvement:

  • Some figure legends and table footnotes require additional clarity, particularly regarding statistical notation.
  • Several sentences are overly long, and the language occasionally becomes redundant; refining the text for precision and clarity would improve readability.
  • There are occasional typographical errors (e.g., "fluor" instead of "flour" in Table 2) and minor formatting issues that need to be addressed.
  • While the statistical methods employed appear appropriate, it is essential that the authors ensure that all comparisons are adequately powered and consider noting any non-significant trends.
  • In certain instances, the distinction between statistically significant group comparisons and non-significant ones is not clear in the tables. More explicit labelling would enhance clarity.
  • The discussion, while adequately relating findings to previous literature, could be streamlined for better flow, and some parts require a clearer explanation of the linkage between oxidative markers and morphological changes, with a more precise statement on the biological implications.

# Analysis by lines:

–Lines 1–17 

  •  Ensure that email addresses and affiliation details maintain consistent formatting (e.g., “santarosa.annecarol@gmail.com” appears twice).
  •  The title is informative but consider shortening it slightly to increase impact.

–Lines 18–20. The opening sentence is slightly convoluted; consider rephrasing to more directly state the problem.

–Line 27. “reduced liver mass” is mentioned but later discussion suggests this is accompanied by increased oxidative stress. Clarify whether the mass reduction correlates directly with the measured biochemical markers.

–Line 30. Instead of “heightened tissue oxidative stress and disrupted hepatic morphology and morphometry,” consider “increased oxidative stress and altered hepatic morphology,” to simplify phrasing.

–Line 36–39. The explanation of gluten components is clear, but the sentence structure could be more concise. For example, “Gluten is composed of prolamins and glutenins; the former provide cohesion and extensibility (e.g., gliadin in wheat) while the latter confers elasticity.”

–Lines 41–46. When discussing peptide formation, add a brief statement on how these peptides might affect liver function to prepare the reader for the hepatic analyses later.

–Lines 57–63. Several sentences review inflammation and oxidative stress; consider integrating these concepts to avoid repetition.  

–Lines 79–83. Clearly state the study rationale. For example, “Given evidence linking gluten to systemic inflammation and oxidative stress, we evaluated the impact of dietary gluten levels on liver morphophysiology in Wistar rats.”

–Lines 91–95. The description of environmental conditions is adequate but verify that temperature is consistently reported (e.g., “22 ± 2°C”).

– Lines 96–102. The grouping is clear; however, in the G70/0 description, “70 days of experimental feeding” could be reworded as “70 days on a high-gluten diet followed by 30 days on a gluten-free diet” for clarity.

– Line 108. The phrase “expanded the mixture” can be simplified as “resulted in pellet expansion.”

– Line 113. “halting further expansion” may be replaced with “preventing additional changes.”

– Line 131. “Wheat fluor” should be corrected to “Wheat flour.”

– In Table 2, review whether the values are consistently presented and check for any typographical misalignments.

–Lines 134–139 (Food Consumption and Body Mass). Ensure that the feed consumption measurement techniques are detailed enough (e.g., measurement intervals) and mention calibration details of the scales.

– Lines 141–142. The description of euthanasia is clear; however, indicate the ethical approval code at the first mention to link with later statement (Line 85).

–Lines 159–169. The MPO assay details in are comprehensive; consider adding a brief rationale for the choice of o-dianisidine concentration.

–Lines 171–193.  Specify the total number of fields or images analyzed per animal if possible.

–Lines 181–182). When describing software (e.g., Image Pro-Plus® include the version number consistently.

–Lines 194–216. The description is clear, but consider a short introductory sentence at Line 194 summarizing why these markers were chosen.

– Lines 202–208. Verify that the notation for DNPH (ε₃₇₀) is consistent with the literature.

–Lines 217–221. Mention that GraphPad Prism is used; ensure that “Prisma® software version 8.0” is correctly cited if that is the intended version. Clarify whether data were checked for normality before applying ANOVA.

–Lines 223–250. Figure 1. Some descriptive phrases (e.g., “potential adaptation period” in Lines 227–229) could be substantiated with numerical data in the figure caption.

–Lines 259–265. Tables 3 and 4 appear detailed. Ensure that the notation for significance (e.g., “a p<0.05 vs. G0; b p<0.05 vs. G14”) is clearly defined in the table footnotes.

–Lines 279–296. In Table 5, consider clarifying the units (for example, “mg/dL” for cholesterol) within the table, and ensure all abbreviations are defined upon first use.

– Line 287. The sentence structure describing MPO activity could be simplified and reorganized for clarity.

– Lines 304–309. The statement on hepatocyte counts is clear, but specify how “significance” was determined (p-values or confidence intervals).

– Lines 317–322: In describing PAS and Sudan III results, ensure that the figure references match the images precisely.

– Recheck that all figure axes are labeled appropriately as referenced in the text.

 – Lines 345–353. The descriptions of carbonylated proteins and TBARS are detailed; consider a summary sentence after presenting these markers to reinforce the overall trend.

– Lines 359–365. When discussing antioxidant enzymes, clarify the relative changes between groups and add comments on potential biological implications.

– Lines 367–373. The introduction to the discussion clearly recapitulates the study rationale; tighten the sentences to remove redundancy.

–Lines 374–386. When referencing previous studies (e.g., Bektaş et al.), provide more context regarding similarities or differences in experimental design.

–Lines 393–400. Clarify the discussion on food consumption results and its lack of correlation with final body mass in some groups.

–Lines 410–417. In the discussion of liver mass, consider referencing specific figures/tables to help the reader correlate textual statements with data.

–Lines 424–438.

  • Explain more clearly the possible confounding effect of rice bran in the G0 diet.
  • Suggest that the non-reference ranges could be discussed in terms of methodological limitations or variability.

–Lines 440–459. The narrative around AST, ALT, and GGT would benefit from a brief discussion of why these markers, despite non-significant differences, are still important to monitor.

– Lines 460–468. When discussing MPO, more clearly delineate whether the increased MPO in certain groups is statistically robust or subject to variability.

– Lines 481–489. The compensatory mechanism proposed is interesting but needs a clearer mechanistic insight. Consider referencing additional literature if available.

– Lines 494–504. The relationship between glycogen content and hepatocyte area is mentioned. Suggest rephrasing for clarity.

–Lines 505–516. This section is detailed and well supported by the data; however, some sentences are too long. Breaking them into shorter, more concise statements will improve clarity.

–Lines 517–525. Ensure that the discussion includes potential reasons for differences observed between catalase and SOD, including any methodological differences or biological variability.

–Lines 533–540. The statement that “removing gluten from the diet could positively reduce oxidative stress” is important. Consider discussing future directions or potential clinical implications.

–Lines 546–554. The conclusions succinctly summarize the main findings. A minor rewording to avoid repetition is recommended, for example, “The NCGS model used here demonstrates that high dietary gluten induces significant changes in liver morphology and oxidative stress markers, indicating potential hepatic implications even in the absence of overt injury.”

# Other issues

–Throughout the manuscript, it is imperative to meticulously review and standardise abbreviations (e.g., NCGS, MPO, TBARS, SOD) at their initial appearance and in subsequent text.

–Typographical errors (e.g., "fluor" for "flour") should be corrected.

–Furthermore, some sentences are excessively long and should be broken into shorter statements to improve readability.

–It is imperative to ensure that all figures and tables are correctly referenced within the text and that the legends provide sufficient detail to facilitate understanding of the data without the need for reference to the text.

–In addition, the incorporation of a concise limitations section would be advantageous, either at the conclusion of the Discussion or as a discrete paragraph, with the objective of acknowledging any inherent limitations of the experimental design, such as the utilisation of an animal model or the consideration of diet composition variables.

Comments on the Quality of English Language

Please, see report

Author Response

# Analysis by lines:

–Lines 1–17 

  •  Ensure that email addresses and affiliation details maintain consistent formatting (e.g., “santarosa.annecarol@gmail.com” appears twice).

Answer: Done.

  •  The title is informative but consider shortening it slightly to increase impact.

Answer: Dear Reviewer,

We appreciate your comment. However, we have chosen to keep our title in this format to emphasize that it refers to a genetically non-gluten-intolerant model. In other words, the title “Non-celiac wheat gluten sensitivity model: effects on hepatic morphophysiology of Wistar rats” aligns with the message we wish to convey to readers.

– Lines 18–20. The opening sentence is slightly convoluted; consider rephrasing to more directly state the problem.

Answer:

Dear Reviewer:  

We believe your comment regarding the sentence was very well taken. Accordingly, we have revised the beginning of our abstract to make it more direct. We have now written: “Wheat gluten intolerance increases intestinal permeability, triggering inflammation that may directly affect liver function and compromise metabolic health”.

– Line 27. “reduced liver mass” is mentioned but later discussion suggests this is accompanied by increased oxidative stress. Clarify whether the mass reduction correlates directly with the measured biochemical markers.

Answer: Dear Reviewer,

We appreciate your comment and agree that it was very pertinent. We correlated the reduction in liver mass with the morphometric and quantitative response of hepatocytes (in the Discussion, lines 509–515) without establishing a relationship with increased oxidative stress. We accept your suggestion and have added a word to the last sentence of the Abstract to improve reader understanding.

– Line 30. Instead of “heightened tissue oxidative stress and disrupted hepatic morphology and morphometry,” consider “increased oxidative stress and altered hepatic morphology,” to simplify phrasing.

Answer: Done.

– Line 36–39. The explanation of gluten components is clear, but the sentence structure could be more concise. For example, “Gluten is composed of prolamins and glutenins; the former provide cohesion and extensibility (e.g., gliadin in wheat) while the latter confers elasticity.”

Answer:

Dear Reviewer,

We have taken your comment regarding the first paragraph of the introduction into consideration and have revised it accordingly. Thank you for the suggestion.

– Lines 41–46. When discussing peptide formation, add a brief statement on how these peptides might affect liver function to prepare the reader for the hepatic analyses later.

Answer:

Dear Reviewer,

We have taken your comment into account and added the following sentence: “These peptides can trigger systemic immune responses and contribute to extra-intestinal manifestations, including alterations in liver function.”. Thank you very much for the observation — it certainly made the text more fluid and easier to read.

– Lines 57–63. Several sentences review inflammation and oxidative stress; consider integrating these concepts to avoid repetition.  

Answer:

Dear Reviewer,

We have accepted your suggestion regarding the paragraph and have revised it accordingly. Thank you for the valuable feedback.

–Lines 79–83. Clearly state the study rationale. For example, “Given evidence linking gluten to systemic inflammation and oxidative stress, we evaluated the impact of dietary gluten levels on liver morphophysiology in Wistar rats.”

Answer:

Dear Reviewer,

We believe your objection was entirely fair, and we have made the suggested change. Thank you.

–Lines 91–95. The description of environmental conditions is adequate but verify that temperature is consistently reported (e.g., “22 ± 2°C”).

Asnwer: Dear Reviewer,

Thank you for your comment. The temperature under observation was determined following the experimental design approved by the ethics committee.

– Lines 96–102. The grouping is clear; however, in the G70/0 description, “70 days of experimental feeding” could be reworded as “70 days on a high-gluten diet followed by 30 days on a gluten-free diet” for clarity.

Answer: Done.

– Line 108. The phrase “expanded the mixture” can be simplified as “resulted in pellet expansion.”

Answer: Done.

– Line 113. “halting further expansion” may be replaced with “preventing additional changes.”

Answer: Done.

– Line 131. “Wheat fluor” should be corrected to “Wheat flour.”

Answer: Done.

– In Table 2, review whether the values are consistently presented and check for any typographical misalignments.

Answer: Dear Reviewer,

We have rectified all typographical misalignments in Table 2. Both the erroneous word and the text that appeared fragmented across two pages have now been properly consolidated. We sincerely appreciate you highlighting these issues.

– Lines 134–139 (Food Consumption and Body Mass). Ensure that the feed consumption measurement techniques are detailed enough (e.g., measurement intervals) and mention calibration details of the scales.

Answer:

Dear Reviewer,

We agree that your observation was very pertinent. Following your suggestion, we have added a sentence at the end of the indicated paragraph to better clarify the analysis details.

– Lines 141–142. The description of euthanasia is clear; however, indicate the ethical approval code at the first mention to link with later statement (Line 85).

Answer: Done.

– Lines 159–169. The MPO assay details in are comprehensive; consider adding a brief rationale for the choice of o-dianisidine concentration.

Answer:

Dear Reviewer,

We appreciate and have incorporated your suggestion. Accordingly, we have revised the text to improve clarity regarding the technique in question.

– Lines 171–193.  Specify the total number of fields or images analyzed per animal if possible.

Answer:

Dear Reviewer,

We appreciate your suggestion. The number of cells, fields, or images used for morphological and morphometric analyses is specified in their respective methodological sections. This represents a well-established morphometric standard routinely employed in our laboratory protocols. However, we found it appropriate to include a reference from our research group.

– Lines 181–182. When describing software (e.g., Image Pro-Plus®) include the version number consistently.

Answer: Done.

– Lines 194–216. The description is clear, but consider a short introductory sentence at Line 194 summarizing why these markers were chosen.

Answer:

Dear Editor,

Your observation was very accurate. Accordingly, we added the necessary explanation to clarify the choice of our markers. Therefore, we included the following sentence: “To comprehensively evaluate hepatic oxidative stress and antioxidant defense mechanisms, specific biochemical markers were selected based on their relevance in detecting protein oxidation, lipid peroxidation, and enzymatic antioxidant activity.”.

– Lines 202–208. Verify that the notation for DNPH (ε₃₇₀) is consistent with the literature.

Answer: Dear Editor,

We have reviewed the DNPH notation and decided to revise it, specifying that 370 is the absorbance value. Thank you very much for your observation.

– Lines 217–221. Mention that GraphPad Prism is used; ensure that “Prisma® software version 8.0” is correctly cited if that is the intended version. Clarify whether data were checked for normality before applying ANOVA.

Answer: Dear Reviewer,

We noted your request for information on how we assessed data normality. A QQ-plot was performed to visually inspect for any deviations from normality. Thank you for this observation; the details have been added to the manuscript (Lines 219–220).

– Lines 223–250. Figure 1. Some descriptive phrases (e.g., “potential adaptation period” in Lines 227–229) could be substantiated with numerical data in the figure caption.

Answer:

Dear Reviewer,

Thank you for your valuable comment. We have incorporated your suggestion and added the information to the text.

– Lines 259–265. Tables 3 and 4 appear detailed. Ensure that the notation for significance (e.g., “a p<0.05 vs. G0; b p<0.05 vs. G14”) is clearly defined in the table footnotes.

Answer:

Dear Reviewer,

We took your concerns into consideration and found them to be very pertinent. Therefore, we adjusted the distribution of significant differences and revised the relevant parts of the footnotes. Thank you very much for your critical insight.

– Lines 279–296. In Table 5, consider clarifying the units (for example, “mg/dL” for cholesterol) within the table, and ensure all abbreviations are defined upon first use.

Answer:

Dear Reviewer,

We appreciate your suggestion. However, we have chosen to retain the units solely in the caption of Table 5 to ensure consistency in presentation. This approach aligns with the format used in Tables 3 and 4.

– Line 287. The sentence structure describing MPO activity could be simplified and reorganized for clarity.

Answer:

Dear Reviewer,

We agree with your observation and have revised the sentence to: “Myeloperoxidase (MPO) activity was significantly increased in the G0 group compared to G14, G42, and G70/0. The G70 group also showed higher MPO activity than the control group (G14) (Table 5)”. Thank you so much.

– Lines 304–309. The statement on hepatocyte counts is clear, but specify how “significance” was determined (p-values or confidence intervals).

Answer: Dear Reviewer,

The p‑value has been duly added. Thank you for your observation.

– Lines 317–322: In describing PAS and Sudan III results, ensure that the figure references match the images precisely.

Answer: Done.

– Recheck that all figure axes are labeled appropriately as referenced in the text.

Answer: Done.

– Lines 345–353. The descriptions of carbonylated proteins and TBARS are detailed; consider a summary sentence after presenting these markers to reinforce the overall trend.

Answer:

Dear Reviewer,

We have taken your observation into account and agree that a sentence summarizing the effects of gluten was missing. Therefore, we have added: “Overall, gluten intake was associated with increased oxidative stress in hepatic tissue, while gluten withdrawal appeared to partially reverse these effects”.

– Lines 359–365. When discussing antioxidant enzymes, clarify the relative changes between groups and add comments on potential biological implications.

Answer:

Dear Reviewer,

New information has been added to the paragraph in question. Your comment was very helpful and contributed to clarifying our idea.

– Lines 367–373. The introduction to the discussion clearly recapitulates the study rationale; tighten the sentences to remove redundancy.

Answer:

Dear Editor,

We have revised the introduction to the discussion and simplified the sentences. It is indeed more cohesive now. Thank you.

– Lines 374–386. When referencing previous studies (e.g., Bektaş et al.), provide more context regarding similarities or differences in experimental design.

Answer:

Dear reviewer,

Thank you for your observation. We have accepted the suggestion and provided more details about the cited works, offering the reader a broader perspective.

– Lines 393–400. Clarify the discussion on food consumption results and its lack of correlation with final body mass in some groups.

Answer:

Dear Reviewer,

We appreciate your comment and have addressed your request accordingly. We have therefore made additions to the text in the specified paragraph.

– Lines 410–417. In the discussion of liver mass, consider referencing specific figures/tables to help the reader correlate textual statements with data.

Answer: Done.

– Lines 424–438.

  • Explain more clearly the possible confounding effect of rice bran in the G0 diet.

Answer:

Dear Reviewer,

Your observation was very accurate. We added a sentence to the paragraph that we believe better explains the negative effect of rice bran on the altered biochemical parameters. Thank you very much.

  • Suggest that the non-reference ranges could be discussed in terms of methodological limitations or variability.

Answer:

Dear Reviewer,

Considering your observation, we have added the limitations of the experiment. Thank you very much. Accordingly, we have included the following sentence: “It is also worth noting that the deviation from reference ranges might reflect methodological limitations or biological variability rather than pathological outcomes alone, and these factors should be considered when interpreting the lipid profile results.”.

– Lines 440–459. The narrative around AST, ALT, and GGT would benefit from a brief discussion of why these markers, despite non-significant differences, are still important to monitor.

Answer:

Dear Reviewer,

Your comment was very important. Discussing these markers and their relevance to liver physiology was essential to further improve our discussion. Therefore, we have added the following paragraph: “Although AST, ALT, and GGT levels did not differ significantly among the ex-perimental groups, these enzymes remain essential markers for monitoring liver health. Subtle, non-significant changes may reflect early or subclinical alterations in hepatic function, especially in experimental contexts involving dietary interventions. The trend toward elevated AST in gluten-fed groups and the variation in ALT levels suggest that gluten intake may influence liver physiology, even in the absence of overt damage [39].”.

– Lines 460–468. When discussing MPO, more clearly delineate whether the increased MPO in certain groups is statistically robust or subject to variability.

Answer:

Dear Reviewer,

Your comment regarding MPO was highly valuable. We revised the paragraph to make our explanation more explicit, leading to a clearer understanding of the result. Thank you.

– Lines 481–489. The compensatory mechanism proposed is interesting but needs a clearer mechanistic insight. Consider referencing additional literature if available.

Answer:

Dear Reviewer,

We sincerely appreciate your suggestion. However, exploring the compensatory mechanisms of the liver falls beyond the current scope of our study. We have therefore chosen to present this as a preliminary hypothesis, as we intend to investigate this compelling aspect more thoroughly in future research through standardized techniques and dedicated publications.

– Lines 494–504. The relationship between glycogen content and hepatocyte area is mentioned. Suggest rephrasing for clarity.

Answer:

Dear Reviewer,

Your comment was taken into consideration, and we have revised the paragraph to make it clearer for the readers. Thank you very much.

– Lines 505–516. This section is detailed and well supported by the data; however, some sentences are too long. Breaking them into shorter, more concise statements will improve clarity.

Answer:

Dear Reviewer,

Your comment was taken into consideration, and we have revised the paragraph to make it clearer for the readers. Thank you very much.

– Lines 517–525. Ensure that the discussion includes potential reasons for differences observed between catalase and SOD, including any methodological differences or biological variability.

Answer:

Dear Reviewer,

The paragraph "The differential behavior between catalase and SOD may reflect distinct regulatory mechanisms and cellular roles for each enzyme. Catalase primarily detoxifies hydrogen peroxide within peroxisomes, whereas SOD dismutates superoxide radicals in both the cytosol and mitochondria. These compartmental and functional differences might result in varying sensitivities to gluten-induced oxidative stress." was added to explain how biological variability may affect the differences observed between catalase and SOD. Thank you so much.

– Lines 533–540. The statement that “removing gluten from the diet could positively reduce oxidative stress” is important. Consider discussing future directions or potential clinical implications.

Answer:

Dear Reviewer,

We appreciate your insightful comment. We agree that the discussion could be strengthened regarding clinical implications, and we have therefore added relevant information on this topic in the final two paragraphs. In these additions, we have emphasized and contextualized the findings by correlating them with existing literature.

– Lines 546–554. The conclusions succinctly summarize the main findings. A minor rewording to avoid repetition is recommended, for example, “The NCGS model used here demonstrates that high dietary gluten induces significant changes in liver morphology and oxidative stress markers, indicating potential hepatic implications even in the absence of overt injury.”

Answer:

Dear Reviewer,

We have accepted your comments regarding the conclusion and believe your observations were very important for improving our work. Thank you very much.

Round 2

Reviewer 2 Report

Comments and Suggestions for Authors

The manuscript has been significantly improved, so it could be published in its current form.